# Inverse LGAD (iLGAD) Periphery Optimization for Surface Damage Irradiation

**DOI:** 10.3390/s23073450

**Published:** 2023-03-25

**Authors:** Albert Doblas, David Flores, Salvador Hidalgo, Neil Moffat, Giulio Pellegrini, David Quirion, Jairo Villegas, Dzmitry Maneuski, Marie Ruat, Pablo Fajardo

**Affiliations:** 1Centro Nacional de Microelectrónica, IMB-CNM-CSIC, 08193 Barcelona, Spain; 2School of Physics and Astronomy, University of Glasgow, Glasgow G12 0YN, UK; 3European Synchrotron Radiation Facility, ESRF, 38000 Grenoble, France

**Keywords:** radiation-hard detectors, fast detectors, X-ray detectors, LGAD, silicon, low energy X-ray

## Abstract

Pixelated LGADs have been established as the baseline technology for timing detectors for the High Granularity Timing Detector (HGTD) and the Endcap Timing Layer (ETL) of the ATLAS and CMS experiments, respectively. The drawback of segmenting an LGAD is the non-gain area present between pixels and the consequent reduction in the fill factor. To overcome this issue, the inverse LGAD (iLGAD) technology has been proposed by IMB-CNM to enhance the fill factor and provide excellent tracking capabilities. In this work, we explore the use of iLGAD sensors for surface damage irradiation by developing a new generation of iLGADs, the periphery of which is optimized to improve the performance of irradiated sensors. The fabricated iLGAD sensors exhibit good electrical performances before and after X-ray irradiation.

## 1. Introduction

Synchrotron facilities are based on charged particles deflected through powerful magnetic fields, producing extremely brilliant light, which wavelength ranges from X-ray to infrared. The photon flux can be up to 10^14^ photons/second with a range of energies between 0.1–200 keV. The ESRF (European Synchrotron Radiation Facility) is one of the most cutting-edge X-ray facilities in Europe with operational energies from 10–120 keV. The ESRF-EBS (Extremely Brilliant Source) upgrade will bring a high-energy light Synchrotron source suitable for a huge variety of ground-breaking research techniques, covering a wide range of applications such as health, biology, climate, energy, and sustainable economy, materials, and innovative industry, world cultural heritage and earth sciences, and planetary research. Radiation detector families are extensively used in X-ray applications, depending on the photon energy. In general, soft X-rays (energy up to 2 keV) are covered by Microchannel Plates (MCPs), Silicon Photodiodes, or Silicon Drift Detectors (SDDs). Hard X-rays up to 20 keV are typically detected with Photomultiplier Tubes (PMT), Avalanche Photodiodes (APD), Hybrid Pixel Detectors (HPD), and indirect detection techniques. Finally, Hard X-rays from 20 keV are commonly detected with CMOS and a-Silicon Flat Panels, High-Z Hybrid Pixel Detectors (HPD), Germanium Detectors, and also indirect detection techniques. In any case, X-ray radiation causes surface damage which is critical in Silicon-based detectors used in energies lower than 20 keV. Therefore, any detector to be used in X-ray applications has to be designed with robust terminations and efficient surface leakage current collection techniques.

Low Gain Avalanche Detectors (LGADs) have been widely studied during the last years by the high energy physics community [1,2,3,4,5]. Based on the Avalanche Photodiode (APD) concept, the LGAD exhibits an intrinsic multiplication in the linear region, providing a gain of 5–20. A p-doped layer, also known as the multiplication layer, is diffused underneath the highly n-doped layer. When a reverse bias is applied a high electric field region is present. The doping level of the multiplication layer is selected to reach a moderate gain, unlike the APD structure. This gain value allows the detector to widen the operating voltage range, at the pixelated electrode, where detection is feasible while maintaining a low signal-to-noise value compared to the APD. A wide operating voltage interval is mandatory since radiation damage will strongly modify the basic device performance. The charge generated in the detector by an incident particle is directly related to its energy. Therefore, the energy of a charged particle going through a detector can be determined by monitoring the generated transient current. In this sense, large-area detectors must be segmented in order to establish the incident point of the particle and to enhance the global precision of the detection measurements. Strip and pixel LGAD sensors, where the multiplication region (n^+^/p) is segmented, are used as tracking detectors and can collect the charge of the incident particle with high precision. However, there is an area between the pixels/strips where the charge does not undergo multiplication (dead area). The fill factor is defined as the ratio of the active area to the total sensor area. In order to reach the maximum fill factor, IMB-CNM has developed the concept of the Inverse Low Gain Avalanche Detector (iLGAD) structure, where the segmentation is performed at the p^+^ electrode of the detector ohmic side and the multiplication region remains unsegmented [6,7]. In the iLGAD structure, the charge generated by the incident particle will be multiplied independently of the hit position. Figure 1 shows a comparison between LGAD and iLGAD structures. Unlike LGADs, the charge collection in the iLGAD is made of holes, which have lower mobility than electrons. Nevertheless, iLGAD shows time resolutions in the range of 20 ps, which is a similar value to that obtained in LGADs [8]. Due to their 100% fill factor, iLGADs are excellent tracking detectors.

The tracking performance at room temperature of one strip iLGAD was bench-marked against one strip LGAD in a test beam at CERN-SPS, using a standard PiN strip detector as a reference [9,10]. The beam particles were pions of 150 GeV with a beam spot of 12 mm. The results are depicted in Figure 2, where two peaks are observed in the LGAD, corresponding to the non-gain (around 24 ke^−^) and the gain regions (77 ke^−^), while in the iLGAD only one peak is present (75 ke^−^) due to the 100 % fill factor [8].

Due to the intrinsic multiplication, low noise (between 3–4 mV in standard thin LGADs [11]), and 100 % fill factor, iLGAD technology is suitable for X-ray detection, where thick detectors are required to collect all the charge generated by X-rays. The Lambert−Beer law states that photons are stopped in a material, according to the material’s absorption, which is a function of energy, thereby decreasing the intensity of the transmitted photon beam [12]. A standard 300 µm silicon detector is able to absorb 13 keV X-rays, but silicon sensors are limited up to 20 keV, as the thickness of the active substrate beyond this energy must be >1 mm. The photoelectric absorption efficiency in a 500 µm thick detector falls to 20% at 25 keV. Therefore, other semiconductors like Cd(Zn)Te or GaAs are under investigation since their attenuation coefficient is much higher than in Silicon [13,14].

The main challenge of developing LGADs for X-ray applications is the radiation damage generated at the Si-SiO_2_ interface. When a device is irradiated electron-hole pairs are created within the interface, and as the electrons have high mobility in silicon they are rapidly collected. However, as holes have lower mobility, a cloud of holes is created inside the SiO_2_. This cloud attracts electrons and an n-doped layer is created underneath the oxide. As a consequence, the electrical performance of the detector is significantly degraded.

In conclusion, the iLGAD detector provides better performance in terms of fill factor than segmented LGADs, but the basic structure has to be optimized for the particular conditions of the X-ray radiation. The periphery optimization is carried out with the aid of Technological Computer-Aided Design (TCAD) tools. The new iLGAD design supports X-ray irradiation, maintaining the electrical performance of the detector. Furthermore, irradiation measurements have been carried out to determine the performance of the new iLGAD detector.

## 2. Optimization of the iLGAD Periphery

The multiplication region is identical in LGAD and iLGAD structures and only the periphery of the detector has to be optimized to make it suitable for synchrotron applications. The first iLGAD prototypes fabricated at IMB-CNM (first iLGAD generation, iLG1) were not designed assuming a high concentration of charges at the Si-SiO_2_ interface, since those devices were not planned to be used as X-ray detectors. The periphery was initially optimized with an n-type junction termination extension (JTE) at the edge of the p-well multiplication layer, introducing a p-type channel stopper at the edge of the device and using highly doped boron diffusion as a collector ring at the ohmic side. An n-type collector ring is diffused at the multiplication side with a distance between the channel stopper and the n-type collector ring sufficiently large to avoid punch-through between both diffusions. Figure 3 shows a schematic view of the complete iLG1 structure.

To simulate the damage created in the detector during X-ray irradiation, a certain oxide charge density (N_ox_) at the Si-SiO_2_ interface has to be considered with a reference value of N_ox_ = 10^12^ cm^−2^, which corresponds to an absorbed radiation dose of Φ = 10 MRad [15]. Figure 4 shows the effect of the charge density on the device breakdown voltage, where a strong reduction of the breakdown voltage is observed when the charge density increases. Thus, the effectiveness of the edge termination is drastically reduced in a harsh X-ray radiation environment. It can be concluded that electrons are accumulated in the oxide-silicon interface and the resulting n-doped layer lowers the electrical performance of the device since it creates a high electric field at the edge of the channel stopper (p-type diffusion). In this situation, the peak electric field is higher than the electric field value created at the n^+^/p junction, leading to a premature breakdown.

To enhance the robustness of the device, the critical electric field peaks must be reduced on both sides of the structure. On the multiplication side, the proposed solution is the addition of five floating rings with their respective p-stop diffusions between them and two more p-stop diffusions between the last floating ring and the channel stopper. As a consequence, a 50 V increase in the breakdown voltage is obtained. To further improve the breakdown voltage capability, the ohmic side of the device has also been optimized. Considering that p-type pixels and the collector ring at the ohmic side are highly-doped diffusions, the electric field created due to the superficial conductive layer is higher than the main junction peak at the multiplication side. Therefore, a field plate contact in the collector ring is introduced to move the electric field peak towards the oxide, increasing the voltage breakdown up to 300 V. Using the same approach as in the multiplication side of the device, the ohmic side includes p-type floating rings and a highly p-type doped channel stopper at the edge. On the ohmic side, n-stop diffusions are not needed since the surface is highly n-doped. Figure 5 shows the electric field distribution along the length of the sensor at the ohmic side (100 nm deep into the Silicon at an applied voltage of 600 V), where the high electric field in the curvature of the diffusions is present. The multi-ring strategy at the multiplication side lowers the peak electric field and increases the breakdown voltage up to 500 V, as shown in Figure 6.

In conclusion, the breakdown voltage has been increased by a factor of four utilizing the optimized design for a harsh X-ray environment, widening the operation voltage regime. Figure 7 shows the final design for the second iLGAD generation (iLG2).

## 3. Fabrication of the Prototypes

The main purpose of the iLGAD structure optimization is to make it suitable for X-ray detection applications. As such, a variety of pixelated sensor layouts have been included in the mask design to be connected to the state-of-the-art readout ASICs for X-ray detectors [16,17]. Therefore, the impact of geometrical parameters on the sensor performance can be studied. Pixelated sensors have 256 × 256 pixels with a pixel pitch of either 55 µm or 75 µm. Additional detectors have been included in the mask design, including strip detectors, to test their performance and the yield of large-area devices. Special emphasis is directed on the correlation of the gain and voltage range with the multiplication layer parameters. Moreover, small pixelated and pad-like detectors with the same periphery as Medipix3 have been included to test their resistance during X-ray irradiation and metal-oxide-semiconductor (MOS) capacitors to study the oxide charges. Finally, LGAD and PiN diodes have been included to control the technological parameters of the process technology such as gain, breakdown voltage, and full depletion voltage (V_FD_), and compare them with the obtained with iLGADs. Test detectors have been fabricated on 4-inch-high resistivity (>1 kΩ · cm) 285 μm thick p-type silicon wafers. Figure 8 shows images of the processed wafers on the multiplication side (left) and the ohmic side (right), corresponding to the second iLGAD generation (iLG2) fabricated at the IMB-CNM clean room. The parameters used for the multiplication region are the same as for standard LGAD detectors since the X-ray irradiation does not affect the performance of the gain layer, contrary to bulk damage caused by proton or neutron irradiation. Medium dose and energy values of the multiplication implant have been selected for the fabrication process.

## 4. Electrical Characterization

iLGAD prototypes have been tested to study their electrical performance and the obtained results have been compared with the previous simulations, to corroborate the good match and the proper tuning of simulator parameters. All the relevant results of the simulated and fabricated sensors are summarized in Figure 9. Figure 9a shows a 1D C-V simulation of an LGAD for different boron doses, selected from previous data of fabricated LGAD sensors. As expected, the depletion of the gain layer increases with the boron dose. Therefore, the gain is also increasing with boron dose, as one can observe in Figure 9b. We expect to have a linear gain between 5–10. As already mentioned, non-pixelated sensors have been added to the mask set in order to test the technology. Figure 9c shows a C-V measurement of a pad-like iLGAD at room temperature where the gain layer depletion voltage (V_GL_) is reached at 38 V and the full depletion at 70 V. The I-V measurement is shown in Figure 9d, showing a leakage current in the range of 10 nA and a breakdown voltage of 450 V, lower than expected, but this value is enough to operate the sensor properly. A metal-oxide-semiconductor (MOS) capacitor is also characterized to extract the oxide charge of an unirradiated sensor. Figure 9e shows the C-V measurement of the MOS capacitor with a flat-band voltage of −6.58 V.

The calculated oxide charge [18] is Qox∼10^11^ cm^−2^, which is the typical value produced after a thermal wet oxidation process. The most critical parameter in LGADs is the gain, which is determined by using the transient current technique (TCT) [19]. The design of the devices includes a window in the aluminum of the multiplication side, in order to allow an infrared (IR) laser to pass through. This measurement was also performed on a pin diode to have a comparison for the gain calculation. Figure 9f shows the gain of the iLGAD, which is calculated as the ratio of the collected charge of an iLGAD and the charge of a pin diode. A linear gain from 12 to 24 is measured in the 70–360 V range, which is higher than expected. In conclusion, the doping concentration of the gain layer is slightly higher than expected, resulting in a higher gain and a lower breakdown voltage. This can be caused by a non-uniformity in the thermal process after the multiplication implantation. Nevertheless, the detector is operative up to 350 V showing a medium linear gain, meeting the requirements of this technology.

## 5. Irradiated Samples

To characterize the endurance of the new radiation-resistant periphery some detectors have been irradiated with X-rays at the X-ray Irradiation facility at the Glasgow Laboratory for Advanced Detector Development (GLADD) using Total Ionising Dose (TID) irradiation experiments. X-rays were generated using a tungsten anode X-ray tube and the dose rate (1.8 MRad/h) was calibrated against a known pre-calibrated PIN diode and verified by the gafchromic film. The devices were attached to a glass plate with acetone-solvable silver epoxy. The glass plate was positioned perpendicular to the X-ray beam at a distance of 20 cm such that X-rays fully cover the devices with dose uniformity over 95 %. The detectors characterized were: a pixelated iLGAD with an active area of 1 × 1 mm^2^ from the second iLGAD generation and a strip iLGAD (8 × 8 mm^2^) from the first iLGAD. Both detectors were irradiated at Φ = 10 MRad at the surface of the detector.

Figure 10 shows the current density of both irradiated and unirradiated detectors, measured in identical conditions. The probes of the probe station are placed in contact with the multiplication region, while the segmented side is in contact with the chuck (ground), short-circuiting the collector ring of the iLG2 design with the pixels and leading to a higher current density in the device. Unfortunately, problems when contacting the pixels have been encountered, as observed in the unirradiated measurements (probes are not properly functioning until 125 V). The iLG2 detector shows a slight increase in the leakage current, but the breakdown voltage capability is not affected. On the other hand, for the iLG1 detector, the voltage capability is completely extinguished, making it of no use for X-ray applications where the sensor bias is in the range of 300 V. Therefore, the irradiation measurements have demonstrated that the sensor has been successfully optimized for X-ray applications, as the performance of the iLG2 detectors is much improved with comparison to those of the first generation. Nevertheless, this statement should be corroborated with a thorough characterization to determine the robustness of the structure to X-ray irradiation.

## 6. Conclusions

The design and development of an iLGAD sensor for X-ray applications are described in this paper. The new structure provides a 100% fill factor, in contrast with the LGAD technology which includes a dead zone between pixels. To use this structure for X-ray detection, a suitable periphery has been designed by means of TCAD tools. The limitations of the first generation against X-ray irradiation are explained and the optimized periphery to achieve a higher operating voltage is proposed. Finally, the ohmic side of the detector has also been optimized to overcome the problems with the high electric field peaks at the curvature of the p^+^ diffusions. The detectors have been fabricated at the IMB-CNM clean room and electrically characterized. Pad-like iLGADs show a lower breakdown voltage and a higher leakage current than expected. This performance is related to the gain, leading to a gain in the range of 12–24, which is slightly higher than the gain obtained by simulation. C-V measurements have been performed to obtain the V_GL_ and V_FD_ of the detectors, in order to establish the range of operation. Moreover, CV measurements have been performed on MOS capacitors to extract the oxide charge of a non-irradiated sensor, which shows the expected result of Q_ox_ ≈ 10^11^ cm^−2^. Finally, some of the fabricated detectors have been irradiated in order to test the new periphery design and compare the I-V behavior between iLGADs of the first and second generations. The optimized design is able to withstand the same voltage before and after irradiation, while the sensor corresponding to the first iLGAD generation cannot be used at a voltage > V_FD_, making it unusable for X-ray applications.

## Figures and Tables

**Figure 1 sensors-23-03450-f001:**
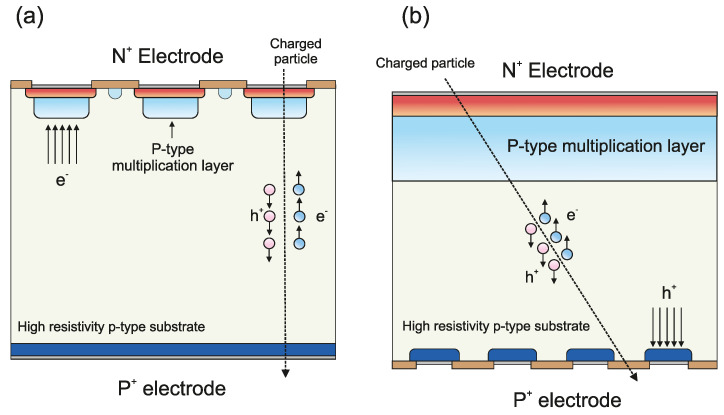
Comparison between (**a**) LGAD and (**b**) iLGAD structures. In the LGAD, the multiplication region is segmented on the n^+^ electrode side, collecting electrons. In the iLGAD, p^+^ electrode is segmented on the ohmic side, collecting holes. A 100 % fill factor is achieved with the iLGAD. The sketch is not to scale.

**Figure 2 sensors-23-03450-f002:**
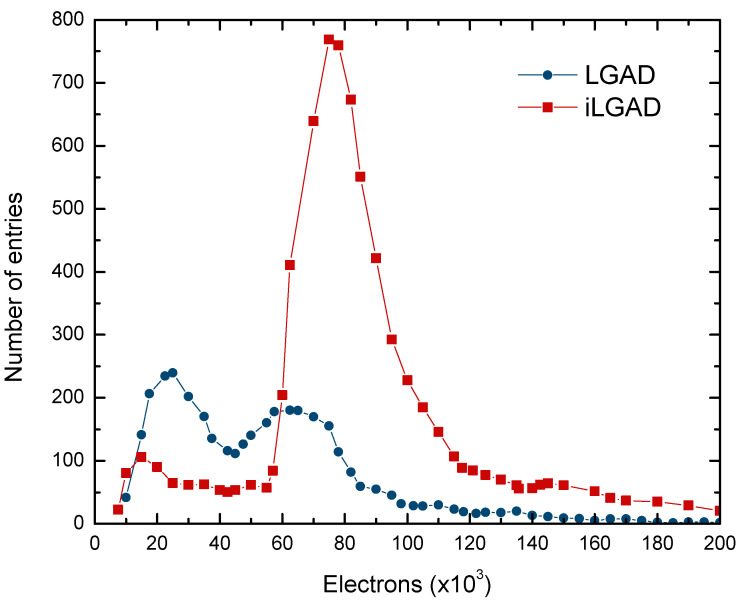
Charge distribution measured during a test beam for one strip LGAD and one strip iLGAD. The iLGAD shows a uniform gain.

**Figure 3 sensors-23-03450-f003:**
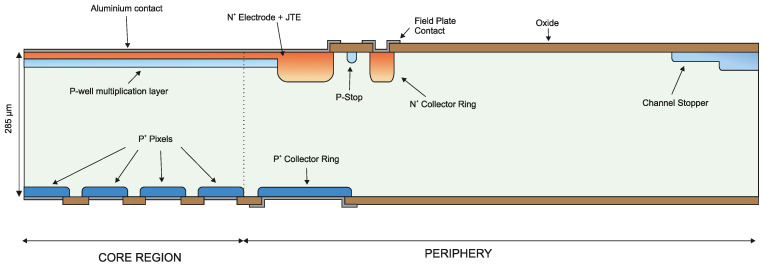
First iLGAD generation (iLG1) fabricated at IMB-CNM, not optimized for X-ray irradiation, with JTE and channel stopper. On the Omhic side, a collector ring is diffused in the periphery.

**Figure 4 sensors-23-03450-f004:**
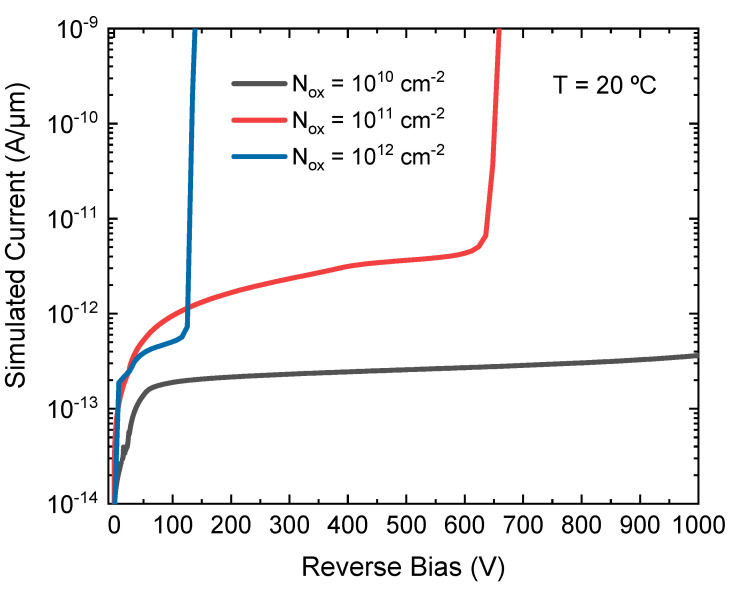
I-V simulation for an iLGAD with different N_ox_ concentrations. Breakdown voltage is reduced by increasing N_ox_.

**Figure 5 sensors-23-03450-f005:**
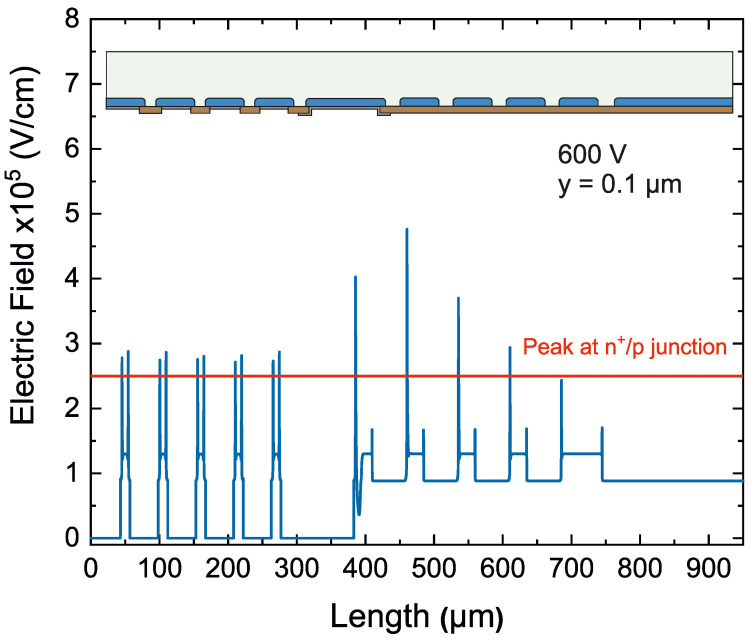
Electric field across the length of the iLGAD. Peaks at the periphery are higher than the maximum peak at the main junction. The conductive electron layer created by the X-rays causes these peaks.

**Figure 6 sensors-23-03450-f006:**
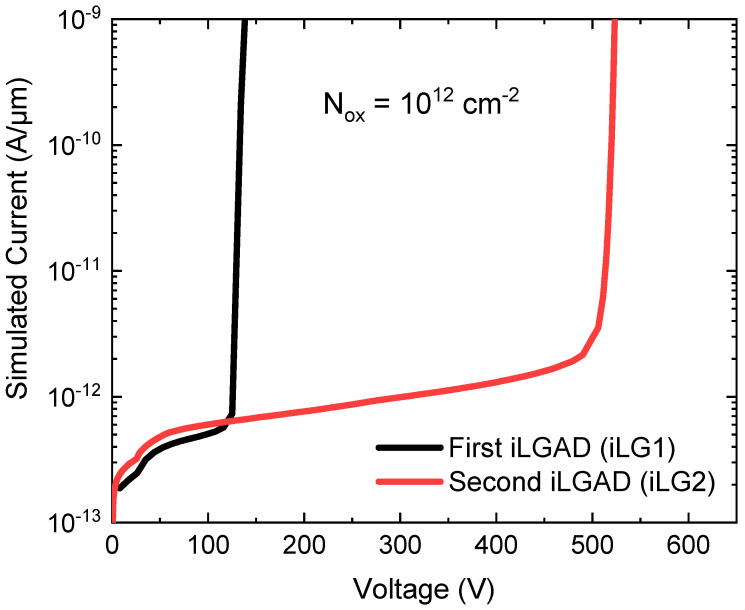
I-V simulation of iLGAD structures with N_ox_ = 10^12^ cm^−2^. The initial and the optimized designs are included to show the significant increase in the breakdown voltage (almost four times).

**Figure 7 sensors-23-03450-f007:**
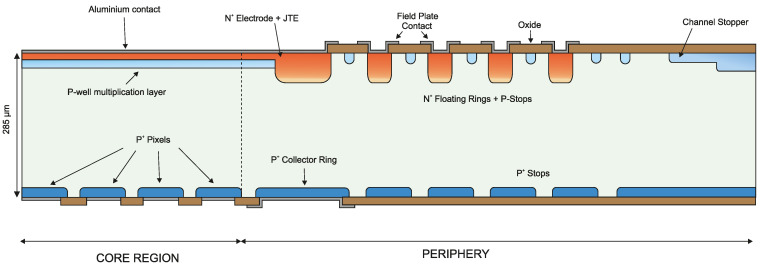
Schematic cross-section of the final periphery design including four n-type floating rings and p-stops at the multiplication side. At the ohmic side, we added four high-doped p-stops and a field plate at the collector ring, to move the high electric field peak towards the oxide.

**Figure 8 sensors-23-03450-f008:**
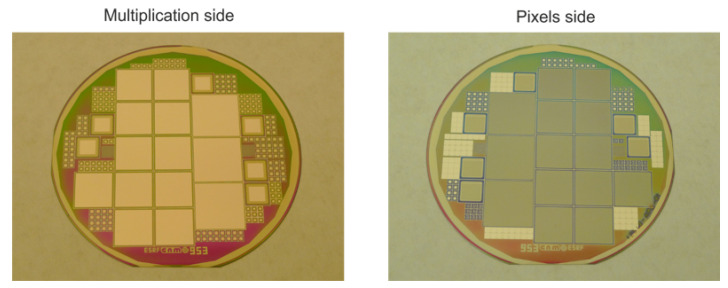
Images of the multiplication and ohmic sides of the fabricated iLGAD wafers.

**Figure 9 sensors-23-03450-f009:**
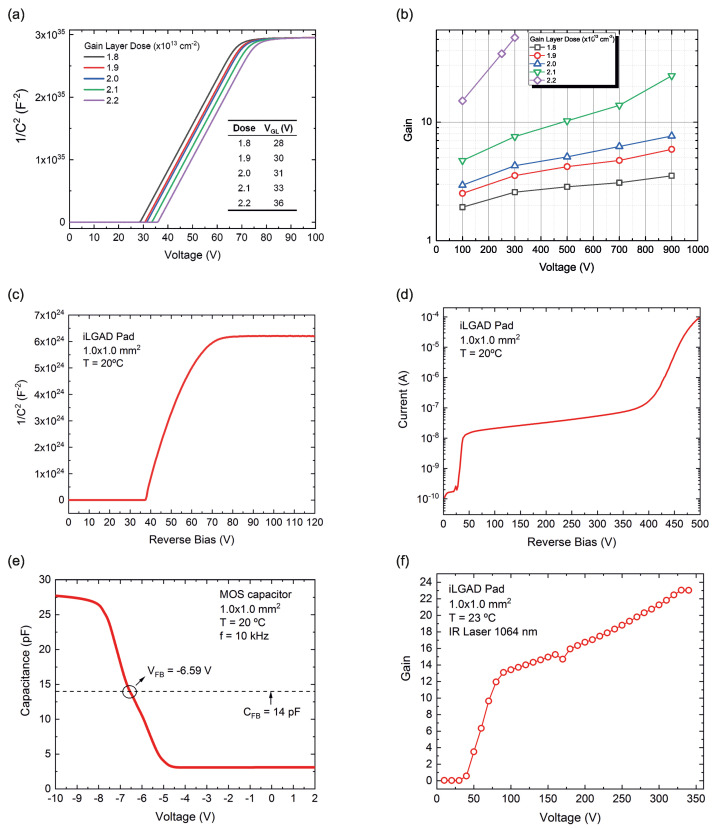
(**a**) C−V simulation for different boron doses. The gain layer increases with increasing boron dose. (**b**) Simulations of the gain for each boron dose. As expected, the gain is increasing with the increase of the boron dose. (**c**) C-V measurement. The gain layer depletion is reached at 38 V. Full depletion voltage is 70 V. (**d**) I-V measurement. Leakage current of 10 nA until the breakdown, which is achieved over 400 V. (**e**) C-V measurement of a MOS capacitor. Flat-band voltage is graphically calculated using the flat-band capacitance value. (**f**) TCT measurement with an IR laser. The gain is calculated as the ratio of the collected charge of an iLGAD and the charge of a pin diode. A linear gain between 12–24 is observed between 70–350 V.

**Figure 10 sensors-23-03450-f010:**
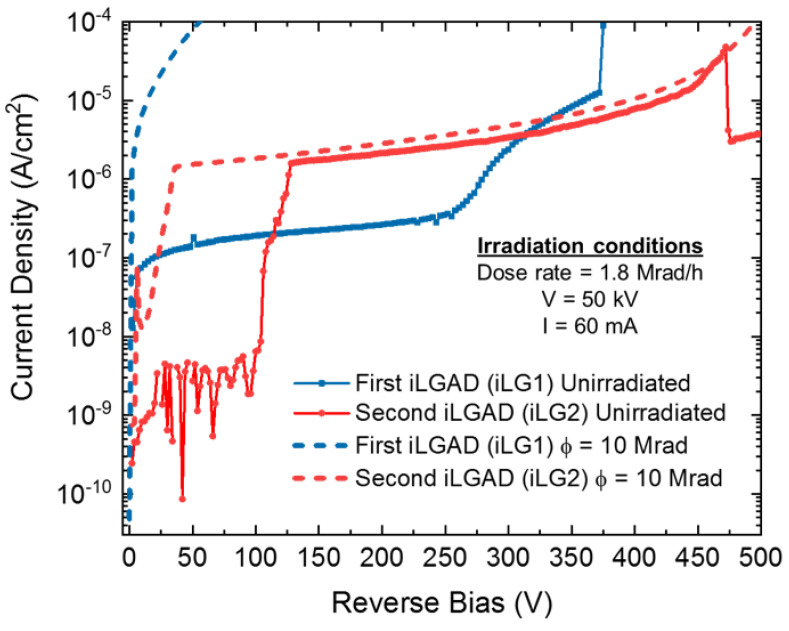
Current density of unirradiated and Φ = 10 MRad irradiated iLGAD detectors from the iLG1 and iLG2 designs.

## Data Availability

Not applicable.

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
