# Peer review of "Inverse LGAD (iLGAD) Periphery Optimization for Surface Damage Irradiation"

_sensors, 2023, doi:10.3390/s23073450_

Round 1

Reviewer 1 Report

While the sensor development effort and measurements described likely have merit, the manuscript needs substantial edits for clarity, accuracy, context, and motivation prior to publication. Detailed comments are in the attached file.

Author Response

The manuscript has been thoroughly revised by the authors. Firstly, English has been revised and improved by a native author. Secondly, all formatting comments have been applied. Finally, all inconsistencies and doubts of the reviewer have been answered. 

In this way, the article now gives a clearer picture of the work done, describing in a better way its objective. Additional technical details on the experimental measurements have been introduced in order to give more information to the reader. In addition, all graphs and figures have been revised and modified. 

Reviewer 2 Report

Review article: Inverted LGAD (iLGAD) Periphery Optimization for Surface Damage Irradiation
Dear Corresponding Author,
the paper is fine.  Few corrections are need:
- the figure caption of fig. 7 is about the same of fig 3.

Author Response

Captions and figures have been reviewed and corrected.

Round 2

Reviewer 1 Report

I appreciate the effort the authors have taken in responding to comments on the first draft. I have additional comments on the revised draft, in the attached document. 

Author Response

Thanks for the comments. Attached you will find the response to your comments.

Round 3

Reviewer 1 Report

While the manuscript has been improved, and the core result (improved tolerance of radiation damage in the new iLGAD design) is solid, I still find that the description of the motivation for development of this technology as an x-ray sensor needs improvement prior to publication. Detailed comments are attached.

Author Response

The point-by-point response is attached
